# An epidemiological study of season of birth, mental health, and neuroimaging in the UK Biobank

Maria Viejo-Romero[1], Heather C. Whalley[1], Xueyi Shen[1], Aleks Stolicyn[1], Daniel J. Smith[1], David M. Howard[1,2]*

1 Division of Psychiatry, Centre for Clinical Brain Sciences, University of Edinburgh, Royal Edinburgh Hospital, Edinburgh, United Kingdom, 2 Institute of Psychiatry, Social, Genetic and Developmental Psychiatry Centre, Psychology & Neuroscience, King's College London, London, United Kingdom

* David.Howard@kcl.ac.uk

**Data Availability Statement:** The data in this study is owned by the UK Biobank (https://www.ukbiobank.ac.uk/) and cannot be publicly shared. All data used in this study, however, can be

## Abstract

Environmental exposures during the perinatal period are known to have a long-term effect on adult physical and mental health. One such influential environmental exposure is the time of year of birth which affects the amount of daylight, nutrients, and viral load that an individual is exposed to within this key developmental period. Here, we investigate associations between season of birth (seasonality), four mental health traits ($n = 137,588$) and multi-modal neuroimaging measures ($n = 33,212$) within the UK Biobank. Summer births were associated with probable recurrent Major Depressive Disorder ($\beta = 0.026$, $p_{corr} = 0.028$) and greater mean cortical thickness in temporal and occipital lobes ($\beta = 0.013$ to $0.014$, $p_{corr} < 0.05$). Winter births were associated with greater white matter integrity globally, in the association fibers, thalamic radiations, and six individual tracts ($\beta = -0.013$ to $-0.022$, $p_{corr} < 0.05$). Results of sensitivity analyses adjusting for birth weight were similar, with an additional association between winter birth and white matter microstructure in the forceps minor and between summer births, greater cingulate thickness and amygdala volume. Further analyses revealed associations between probable depressive phenotypes and a range of neuroimaging measures but a paucity of interactions with seasonality. Our results suggest that seasonality of birth may affect later-life brain structure and play a role in lifetime recurrent Major Depressive Disorder. Due to the small effect sizes observed, and the lack of associations with other mental health traits, further research is required to validate birth season effects in the context of different latitudes, and by co-examining genetic and epigenetic measures to reveal informative biological pathways.

## Introduction

Season of birth has long been hypothesised to have enduring effects on human health [1]. The relatively recent "Foetal Origins of Adult Disease" hypothesis proposes that intra-uterine exposures influence later adult health, including mental health outcomes [2,3]. Seasonality of birth

accessed upon application through the UK Biobank Access Management System (https://ams. ukbiobank.ac.uk/ams/). All permissibly shared data for this study can be found in the Supplementary Information file.

**Funding:** M.V.R. is supported under a 2018 NARSAD Young Investigator Grant from the Brain & Behavior Research Foundation (Ref: 27404). D. M.H. is supported by a Sir Henry Wellcome Postdoctoral Fellowship (Reference 213674/Z/18/ Z) and a 2018 NARSAD Young Investigator Grant from the Brain & Behavior Research Foundation (Ref: 27404). STRADL UKB Application (#4844) was funded by the Wellcome Trust (Ref: 104036/Z/ 14/Z). This research was funded in whole, or in part, by the Wellcome Trust [Reference 213674/Z/ 18/Z]. For the purpose of open access, the author has applied a CC BY public copyright licence to any Author Accepted Manuscript version arising from this submission. The funders had no role in study design, data collection and analysis, decision to publish, or preparation of the manuscript.

**Competing interests:** The authors have declared that no competing interests exist

has been associated with multiple diseases [4], including psychiatric [5–7], neurodevelopmental [8], cardiovascular [9], inflammatory [10], and genetic [11]. The mechanisms by which seasonality may affect risk to these disorders are hypothesised to include interactions with photoperiod and sunlight [12], nutrition [13], risk of pre-term birth and early life infection [14], and maternal vitamin D deficiency [15]. These interactions could also be mediated by genetic [16], epigenetic [17] or environmental changes [5] *in utero* or perinatally.

Perinatal photoperiod directly affects the physiology, brain morphology and behaviour of many animals via transplacental signalling to melatonin receptors in the developing medio-basal hypothalamus [18,19]. Many of these season-of-birth effects endure into adulthood, including altered circadian timing [20], changes in affective behaviour, hippocampal volume [21] and changes to serotonergic and dopaminergic systems in the brain [22]. The wider effects of season of birth on human brain structure have been investigated using magnetic resonance imaging with winter births associated with increased grey matter volume of the superior temporal gyrus in a study of over 550 individuals [23]. However, in a larger study of 13,000 Rotterdam Study participants there was no effect of season of birth on any imaging derived measures [24].

Differences in photoperiod exposure during the perinatal and postweaning time windows have been associated with enduring changes in anxiety and depressive-like behaviours in developmentally stable adulthood periods within animal models [25,26], with shorter photoperiods associated with increased presence and severity of these behaviours. Differences in brain morphology have been associated with mental health disorders including depression [27–29], externalizing behaviour [30] and schizophrenia [31,32]. The associations between brain morphology and mental health thus warrant an investigation of associations of seasonality with both brain morphology and mental health.

Birth seasonality potentially plays a multifaceted role in the aetiology of adult mental health, either alongside or independently of seasonality-induced brain morphology changes. However, to date, the relationship between seasonality of birth, mental health traits and imaging measures has not been examined at scale. This study, therefore, aims to explore associations between seasonality of birth and (a) mental health disorders, and (b) brain imaging measures within the UK Biobank, while also examining the potential interaction between mental health disorders and seasonality of birth on brain imaging measures.

## Materials & methods

### UK Biobank

The UK Biobank (UKB) is a well-characterised community cohort of over 500,000 participants aged 37–73 years at recruitment (2006–2010) [33]. All participants were invited to an initial assessment in which baseline data were collected, including month of birth, birth weight, and birth location. Mental health information was collected for all individuals at baseline using a Touchscreen Questionnaire, and for 157,348 participants between 2016 and 2017 using a Mental Health Questionnaire [34] (MHQ). A subset of the cohort ($n$ = 42,709) was invited to the first imaging visit in 2014 where brain scans, including T1-weighted (T1) and diffusion tensor imaging (DTI), were obtained. Data were accessed under the UKB Application Number 4844. UKB has approval from the NHS National Research Ethics Service as a research tissue bank (References 16/NW/0274 and 11/NW/0382).

### Seasonality

Seasonality of birth was examined as a quantitative phenotypic trait ($y$) capturing the month of birth (UKB data-field 52) following the approach of Howard et al. [35]. The birth month of each participant ($i$) was transformed via a cos function, with the lowest phenotypic score (-1)

corresponding to those born in December and the highest phenotypic score corresponding to those born in June (+1), as below:

$$y_i = -1 \times cos\left(2\pi\left(\frac{Month\ of\ Birth_i}{12}\right)\right)$$

## Mental health traits

Four mental health trait phenotypes were obtained from responses to the Touchscreen Questionnaire and the "Thoughts and Feelings" section of the MHQ. Where participants had completed both questionnaires ($n = 85,266$), the MHQ responses were used to generate phenotypes due to the greater specificity of this questionnaire (see S2.2.1A and S2.2.1B Methods in S1 File for UKB data-fields used). Participants who had answered "Prefer not to answer" or "I don't know" to any questions or with insufficient symptom data (i.e., only one reported symptom) were removed at this stage. The remaining participants ($n = 170,982$) were categorised into four mental health-related phenotypes: probable recurrent Major Depressive Disorder (P-RMDD) ($n = 39,528$ cases), probable single episode Major Depressive Disorder (P-SEMDD) ($n = 16,430$ cases), probable Hypomania ($n = 9,104$ cases) and probable Mania ($n = 2,569$ cases), plus a control group ($n = 103,351$) as previously validated by Smith et al. [36] (see S2.2.1C and S2.2.1D Methods in S1 File for grouping specification). At this point, participants who did not report a UK or Republic of Ireland birthplace or had unrecoverable geographical birthplace co-ordinate data ($n = 16,946$) (See S2.1 Methods in S1 File for geographical data processing), had missing Townsend Deprivation Index values ($n = 192$), or had chosen to withdraw from further studies ($n = 9$) were excluded. Furthermore, participants who reported other neuropsychiatric or sleep-related conditions ($n = 28$), brain cancer diagnoses ($n = 2$), or reported having done shift work ($n = 13,631$) were also excluded. This resulted in a total of 30,808 exclusions with 140,174 participants remaining (see S2.3.1A Methods in S1 File for exclusion details).

Any overlaps between the groupings were removed to create distinct phenotypes that did not share a probable diagnosis: P-RMDD ($n = 32,285$), P-SEMDD ($n = 13,721$), probable Unipolar Mania (P-UM) ($n = 1,229$) and probable Bipolar Depression (P-BD) ($n = 5,278$) (see S2.2.1D Methods in S1 File for overlaps permitted). Both the P-RMDD and the P-SEMDD groups exclude participants within the probable Mania or probable Hypomania groups. P-UM excludes participants within either depression group, whereas the P-BD grouping allows for participants also within either depression grouping and follows the definitions used in Sangha et al. [37]. Participants who completed the MHQ lead questions for mania (UKB data-fields 20501 and 20502) and depression (UKB data-fields 20446, 20441), and/or the Touchscreen Questionnaire lead questions for mania (UKB data-fields 4642, 4653) and depression (UKB data-fields 4598, 4631) but had not been classified into one of the four phenotypes, were defined as controls ($n = 84,018$). The total sample size was therefore 137,588 (see S2.3.1B Methods in S1 File for group sample sizes). Participant demographics for the mental health phenotypes are provided in Table 1.

## Brain imaging measures

T1-weighted and DTI brain scans were obtained for 42,709 participants on their first imaging visit [38], with complete brain imaging data available for 37,048 participants. Participants with outlier values in global measures of cortical surface area ($n = 112$), mean cortical thickness ($n = 217$), cortical volume ($n = 117$), subcortical volume ($n = 90$), mean diffusivity (MD)

**Table 1. Participant demographic details for mental health trait analysis.**

| | Control (N = 85075) | Probable Major Depressive Disorder (N = 13721) | Probable Single Episode Depression (N = 32285) | Probable Unipolar Mania (N = 1229) | Probably Bipolar Depression (N = 5278) | Total (N = 137588) |
|---|---|---|---|---|---|---|
| **Sex** | | | | | | |
| Female | 39772 (46.7%) | 9433 (68.7%) | 21765 (67.4%) | 474 (38.6%) | 3132 (59.3%) | 74576 (54.2%) |
| Male | 45303 (53.3%) | 4288 (31.3%) | 10520 (32.6%) | 755 (61.4%) | 2146 (40.7%) | 63012 (45.8%) |
| **Age (years)** | | | | | | |
| Mean (SD) | 62.7 (8.40) | 61.4 (8.12) | 60.2 (8.33) | 61.1 (8.53) | 59.7 (7.85) | 61.9 (8.42) |
| Median [Min, Max] | 64.0 [40.2, 80.5] | 62.2 [40.3, 79.2] | 60.9 [40.2, 79.6] | 61.9 [40.4, 79.4] | 59.4 [40.3, 80.3] | 62.9 [40.2, 80.5] |
| **Ethnicity** | | | | | | |
| Prefer not to answer | 177 (0.2%) | 28 (0.2%) | 83 (0.3%) | 3 (0.2%) | 12 (0.2%) | 303 (0.2%) |
| Do not know | 8 (0.0%) | 4 (0.0%) | 6 (0.0%) | 0 (0%) | 0 (0%) | 18 (0.0%) |
| White | 62 (0.1%) | 7 (0.1%) | 29 (0.1%) | 2 (0.2%) | 8 (0.2%) | 108 (0.1%) |
| Mixed | 3 (0.0%) | 1 (0.0%) | 4 (0.0%) | 0 (0%) | 1 (0.0%) | 9 (0.0%) |
| Asian or Asian British | 1 (0.0%) | 0 (0%) | 1 (0.0%) | 0 (0%) | 0 (0%) | 2 (0.0%) |
| Black or Black British | 3 (0.0%) | 0 (0%) | 0 (0%) | 0 (0%) | 0 (0%) | 3 (0.0%) |
| Chinese | 18 (0.0%) | 4 (0.0%) | 5 (0.0%) | 0 (0%) | 1 (0.0%) | 28 (0.0%) |
| Other ethnic group | 89 (0.1%) | 31 (0.2%) | 85 (0.3%) | 3 (0.2%) | 19 (0.4%) | 227 (0.2%) |
| British | 81892 (96.3%) | 13129 (95.7%) | 30528 (94.6%) | 1152 (93.7%) | 4909 (93.0%) | 131610 (95.7%) |
| Irish | 1221 (1.4%) | 225 (1.6%) | 590 (1.8%) | 23 (1.9%) | 131 (2.5%) | 2190 (1.6%) |
| Any other white background | 834 (1.0%) | 168 (1.2%) | 510 (1.6%) | 25 (2.0%) | 93 (1.8%) | 1630 (1.2%) |
| White and Black Caribbean | 73 (0.1%) | 16 (0.1%) | 35 (0.1%) | 1 (0.1%) | 13 (0.2%) | 138 (0.1%) |
| White and Black African | 24 (0.0%) | 6 (0.0%) | 29 (0.1%) | 4 (0.3%) | 4 (0.1%) | 67 (0.0%) |
| White and Asian | 75 (0.1%) | 16 (0.1%) | 61 (0.2%) | 1 (0.1%) | 23 (0.4%) | 176 (0.1%) |
| Any other mixed background | 73 (0.1%) | 16 (0.1%) | 61 (0.2%) | 2 (0.2%) | 18 (0.3%) | 170 (0.1%) |
| Indian | 116 (0.1%) | 11 (0.1%) | 55 (0.2%) | 6 (0.5%) | 9 (0.2%) | 197 (0.1%) |
| Pakistani | 37 (0.0%) | 4 (0.0%) | 23 (0.1%) | 2 (0.2%) | 6 (0.1%) | 72 (0.1%) |
| Bangladeshi | 3 (0.0%) | 0 (0%) | 2 (0.0%) | 0 (0%) | 0 (0%) | 5 (0.0%) |
| Any other Asian background | 13 (0.0%) | 2 (0.0%) | 6 (0.0%) | 0 (0%) | 3 (0.1%) | 24 (0.0%) |
| Caribbean | 292 (0.3%) | 44 (0.3%) | 141 (0.4%) | 5 (0.4%) | 19 (0.4%) | 501 (0.4%) |
| African | 58 (0.1%) | 7 (0.1%) | 23 (0.1%) | 0 (0%) | 6 (0.1%) | 94 (0.1%) |
| Any other Black background | 3 (0.0%) | 2 (0.0%) | 8 (0.0%) | 0 (0%) | 3 (0.1%) | 16 (0.0%) |
| **Month of Birth** | | | | | | |
| January | 6987 (8.2%) | 1152 (8.4%) | 2720 (8.4%) | 110 (9.0%) | 454 (8.6%) | 11423 (8.3%) |
| February | 6785 (8.0%) | 1099 (8.0%) | 2518 (7.8%) | 95 (7.7%) | 390 (7.4%) | 10887 (7.9%) |
| March | 7727 (9.1%) | 1256 (9.2%) | 2989 (9.3%) | 112 (9.1%) | 460 (8.7%) | 12544 (9.1%) |
| April | 7372 (8.7%) | 1191 (8.7%) | 2858 (8.9%) | 125 (10.2%) | 446 (8.5%) | 11992 (8.7%) |
| May | 7664 (9.0%) | 1221 (8.9%) | 2961 (9.2%) | 103 (8.4%) | 464 (8.8%) | 12413 (9.0%) |
| June | 7016 (8.2%) | 1151 (8.4%) | 2852 (8.8%) | 113 (9.2%) | 454 (8.6%) | 11586 (8.4%) |
| July | 7251 (8.5%) | 1161 (8.5%) | 2741 (8.5%) | 115 (9.4%) | 452 (8.6%) | 11720 (8.5%) |
| August | 7011 (8.2%) | 1186 (8.6%) | 2588 (8.0%) | 99 (8.1%) | 443 (8.4%) | 11327 (8.2%) |
| September | 7010 (8.2%) | 1139 (8.3%) | 2581 (8.0%) | 93 (7.6%) | 474 (9.0%) | 11297 (8.2%) |
| October | 6818 (8.0%) | 1096 (8.0%) | 2518 (7.8%) | 92 (7.5%) | 425 (8.1%) | 10949 (8.0%) |

*(Continued)*

**Table 1.** (Continued)

| | Control (N = 85075) | Probable Major Depressive Disorder (N = 13721) | Probable Single Episode Depression (N = 32285) | Probable Unipolar Mania (N = 1229) | Probably Bipolar Depression (N = 5278) | Total (N = 137588) |
|---|---|---|---|---|---|---|
| November | 6567 (7.7%) | 999 (7.3%) | 2420 (7.5%) | 92 (7.5%) | 394 (7.5%) | 10472 (7.6%) |
| December | 6867 (8.1%) | 1070 (7.8%) | 2539 (7.9%) | 80 (6.5%) | 422 (8.0%) | 10978 (8.0%) |
| **Birth Location Cluster** | | | | | | |
| 1 | 5707 (6.7%) | 932 (6.8%) | 1958 (6.1%) | 111 (9.0%) | 395 (7.5%) | 9103 (6.6%) |
| 2 | 49494 (58.2%) | 7964 (58.0%) | 18786 (58.2%) | 683 (55.6%) | 2977 (56.4%) | 79904 (58.1%) |
| 3 | 8727 (10.3%) | 1466 (10.7%) | 3283 (10.2%) | 115 (9.4%) | 559 (10.6%) | 14150 (10.3%) |
| 4 | 21147 (24.9%) | 3359 (24.5%) | 8258 (25.6%) | 320 (26.0%) | 1347 (25.5%) | 34431 (25.0%) |

SD = Standard deviation. Birth location was obtained using k-means clustering (see Statistical models section for full details).

($n$ = 105), or fractional anisotropy (FA) ($n$ = 232) were excluded. Global measures were derived by conducting principal component analyses (PCA) on data from the entire sample [39], and measure outliers were defined as values ±3 standard deviations from the sample mean for that measure. Participants with non-UK/Republic of Ireland, non-specified ($n$ = 2,635) or unrecoverable geographical ($n$ = 590) birthplace, with a missing Townsend Deprivation Index value ($n$ = 26), or who wished to withdraw from future studies ($n$ = 3) were also excluded. No participants had to be removed due to head motion in the scanner. A total of 33,212 participants were included after 3,254 exclusions. Demographics of participants included in the analyses of brain imaging measures are provided in Table 2. An additional sensitivity analysis was conducted using only individuals with birth weight data ($n$ = 21,417).

**Brain morphology measures.** Brain morphology measures were obtained by the UK Biobank with FreeSurfer 6.0 toolkit [40–42] and included volumes of seven subcortical structures (SV) for each hemisphere, as well as cortical volume (CV), mean cortical thickness (CT) and cortical surface area (CSA) measures of 31 cortical regions for each hemisphere, based on the Desikan-Killiany-Tourville atlas [43] ($n$ = 33,212, see S1.1 Methods in S1 File for full pre-analysis QC). Similar measures were analysed in Harris et al. [39] (UKB category 192) and Shen et al. [29]. Global and lobar CT, CSA and CV measures were derived manually for each hemisphere (n = 33,212). Lobar measures were obtained for the frontal, parietal, temporal, occipital and cingulate lobes (see S2.2.2A Methods in S1 File for group composition). All brain morphometric measures were normalised.

**White matter microstructure measures.** White matter (WM) microstructure measures consisted of FA and MD values for 12 bilateral tracts and 3 unilateral tracts derived by the UK Biobank with the FSL probabilistic tractography toolkit (See S1.1 Methods in S1 File) [38,44]. Additional fiber-related FA and MD measures were derived as the scores on the first unrotated principal components from PCA, which combined bi-hemispheric (left and right) and unilateral measures from all relevant individual fiber tracts. The three whole-brain fiber bundles derived were the association fibers (gAF), projection fibers (gPF) and thalamic radiations (gTR) (see Methods S2.2.2A in S1 File for fiber definitions). Global FA (gFA) and MD (gMD) measures were derived as the scores on the first unrotated principal components from PCA analyses which combined all bi-hemispheric and unilateral tract measures. Proportions of variance explained by the first principal components are provided in Methods S2.2.2B in S1 File. Thirty-eight WM integrity measures were analysed in total (19 FA and 19 MD) ($n$ = 33,212) and all WM microstructure measures were normalised.

**Table 2. Participant demographic details for neuroimaging analysis.**

| | Total (N = 33212) |
|---|---|
| **Sex** | |
| Female | 17731 (53.4%) |
| Male | 15481 (46.6%) |
| **Age (years)** | |
| Mean (SD) | 63.6 (7.45) |
| Median [Min, Max] | 64.0 [45.0, 82.0] |
| **Ethnicity** | |
| Prefer not to answer | 73 (0.2%) |
| Do not know | 5 (0.0%) |
| White | 16 (0.0%) |
| Mixed | 1 (0.0%) |
| Asian or Asian British | 0 (0%) |
| Black or Black British | 0 (0%) |
| Chinese | 10 (0.0%) |
| Other ethnic group | 56 (0.2%) |
| British | 31812 (95.8%) |
| Irish | 578 (1.7%) |
| Any other white background | 409 (1.2%) |
| White and Black Caribbean | 31 (0.1%) |
| White and Black African | 12 (0.0%) |
| White and Asian | 40 (0.1%) |
| Any other mixed background | 32 (0.1%) |
| Indian | 44 (0.1%) |
| Pakistani | 11 (0.0%) |
| Bangladeshi | 2 (0.0%) |
| Any other Asian background | 4 (0.0%) |
| Caribbean | 61 (0.2%) |
| African | 14 (0.0%) |
| Any other Black background | 1 (0.0%) |
| **Birth Weight (kg)** | |
| Mean (SD) | 3.36 (0.614) |
| Median [Min, Max] | 3.37 [0.740, 6.78] |
| Missing | 11795 (35.5%) |
| **Month of Birth** | |
| January | 2763 (8.3%) |
| February | 2701 (8.1%) |
| March | 3056 (9.2%) |
| April | 2906 (8.7%) |
| May | 2905 (8.7%) |
| June | 2789 (8.4%) |
| July | 2838 (8.5%) |
| August | 2744 (8.3%) |
| September | 2698 (8.1%) |
| October | 2669 (8.0%) |
| November | 2475 (7.5%) |
| December | 2668 (8.0%) |

(*Continued*)

**Table 2.** (Continued)

| | Total (N = 33212) |
|---|---|
| **Birth Location Cluster** | |
| 1 | 2658 (8.0%) |
| 2 | 12037 (36.2%) |
| 3 | 6175 (18.6%) |
| 4 | 12342 (37.2%) |
| **Scanner Site** | |
| Cheadle | 20377 (61.4%) |
| Reading | 4042 (12.2%) |
| Newcastle | 8793 (26.5%) |

SD = Standard deviation. Birth location was obtained using k-means clustering (see Statistical models section for full details).

## Statistical models

**Mental health traits.** To investigate associations between the four mental health phenotypes and seasonality, logistic binomial regression analyses were performed, covarying for sex, age, $age^2$, Townsend Deprivation Index (TDI), assessment centre attended and place of birth location. Place of birth locations were derived with k-means clustering of participant birth north / east co-ordinates from the Ordnance Survey data (UKB variables 129 and 130), performed using "kclust" function in the R "stats" package. Twenty clustering iterations were run to identify four birth location clusters (see S2.1.1 Methods and S1-S3 Fig in S1 File), and each participant was assigned to a cluster to define place of birth. TDI was included as a covariate to capture sociodemographic factors known to be associated with psychiatric disorders in UK Biobank [45,46]. The inclusion of place of birth as a covariate was based on previous associations between birth location and genetic variants in UK Biobank [47], as well as findings of associations between higher latitude births and lifetime risk for depression [48]. $Age^2$ was included to account for the non-linear relationship between age and mental health symptoms, namely for depressive symptoms [49], especially when modelling early-life events [50]. Sex, assessment centre and place of birth were coded as categorical variables. A Bonferroni multiple analysis correction was applied over the four phenotypes examined (P < 0.0125 ($\alpha$ = 0.05 / 4)). Effect sizes were standardised throughout. No other statistical models were tested in this study. Post-hoc analysis was conducted to test the robustness of this model by reducing the number of covariates to sex, age and assessment centre attended and sequentially adding birth location, TDI and $age^2$ as covariates to further binomial logistic regressions per mental health trait (n = 16) (i.e., covariates = age, sex, assessment centre; age, sex, assessment centre, birth location; age, sex, assessment centre, birth location, TDI; age, sex, assessment centre, birth location, TDI, $age^2$). A Bonferroni multiple analysis correction was applied consistent with that applied in the main study.

**Brain imaging measures.** Linear regression models were applied to assess associations between seasonality and all unilateral, fiber-related or global brain measures. Mixed-effects models were applied to assess associations between seasonality and all bilateral brain measures ("nlme" package in R version 3.2.3). Sex, age, $age^2$, Townsend Deprivation Index, assessment centre, four UKB head position covariates (X, Y, Z and table position) and place of birth cluster index (see S4-S6 Fig in S1 File) were included as covariates in all analyses. The rationale for these covariates is similar to that of the mental health traits models with TDI included due to previous

associations with brain structure in this cohort [51], age$^2$ to account for non-linearity in the relationship between age and brain imaging measures [29], and birth place given its associations with genetic variants also within this cohort [47]. Sex, assessment centre and place of birth were coded as categorical variables. Hemisphere was controlled for as a random effect in the mixed-effect models for all bi-hemispheric measures (see S2.1.2 Methods in S1 File). Standardised intracranial volume was also covaried for in all analyses of brain morphometric measures.

False Discovery Rate (FDR) correction was applied separately across individual and regional brain morphology and white matter microstructure measures per modality (Methods S2.4.1 in S1 File) [29,39]. Global measures were not corrected. A P-value threshold for significance was set to 0.05 and effect sizes were standardised throughout. All statistical analyses were performed using R (version 3.2.3). A sensitivity analysis was conducted to reassess the brain imaging measures for an association with seasonality after fitting birth weight as an additional fixed effect covariate.

Further analyses were conducted to explore associations between brain imaging measures and additional mental health phenotypes. First, models to examine associations between brain imaging measures and probable Major Depressive Disorder cases (P-MDD) were constructed. P-MDD included all participants who had been previously classified as P-RMDD or P-SEMDD cases, with P-MDD controls defined as those participants who were controls for both P-RMDD or P-SEMDD (See S2.5.1 in S1 File for demographic table). Secondly, models were constructed to explore associations between brain imaging measures and winter birth P-MDD cases compared to summer birth P-MDD cases. Seasonality was split into winter births (December, January, February) and summer births (June, July, August) and coded as a categorical variable (See S2.5.2 in S1 File for demographic table). Thirdly, additional models were constructed to investigate associations between brain imaging measures and P-RMDD cases compared to P-SEMDD cases (See S2.5.3 in S1 File for demographic table). All covariates were kept consistent with the main analysis. Lastly, models were constructed to investigate associations between mental health traits and neuroimaging measures in the context of seasonality. For each neuroimaging measure a linear/fixed effect model was applied, retaining the same covariates as in the main analysis plus one mental health trait (n = 4: P-RMDD, P-SEMDD, P-BD and P-UM) and an interaction term between the mental health trait and seasonality. As in the main analysis, mixed-effects models were used for bi-hemispheric measures. The same multiple correction strategy was applied as in the main analysis. No other statistical models were tested in this study.

## Results

### Seasonality associations with mental health traits

P-RMDD was associated with seasonality, with a higher prevalence observed in summer births ($\beta$ = 0.026, $p_{corr}$ = 0.028). No other mental health traits were associated with seasonality (Table 3). Effect sizes are reported as log-transformed odd ratios.

**Table 3. Mental health traits associated with seasonality.**

| Mental Health Trait | Effect Size(β) / Log(OR) | S.E. | p-uncorr | p-corr |
|---|---|---|---|---|
| Probable recurrent Major Depressive Disorder | 0.026 | 0.010 | 0.007 | 0.028 |
| Probable single Episode Major Depressive Disorder | 0.017 | 0.013 | 0.212 | 0.847 |
| Probable Unipolar Mania | 0.066 | 0.041 | 0.108 | 0.432 |
| Probable Bipolar Depression | 0.009 | 0.020 | 0.665 | 1.000 |

p-uncorr = p-uncorrected value; p-corr = Bonferroni p-corrected value; S.E = standard error.

Post-hoc robustness analysis for mental health trait associations with seasonality revealed a consistent pattern of associations across all models tested. P-RMDD was significantly associated with summer births across all models tested *(n* = 4; β range: 0.024–0.026, $p_{corr}$ <0.05), with the largest effect size observed within the model utilised in the main analysis (β = 0.026) (see S3.3 Results in S1 File).

### Seasonality associations with brain imaging measures

Greater temporal lobe CT (β = 0.014, $p_{corr}$ = 0.037) and greater occipital lobe CT (β = 0.013, $p_{corr}$ = 0.037) were associated with summer births (Fig 1).

Winter births were associated with higher gFA (β = -0.017, *p* = 0.001), higher gAF and gTR (respectively β = -0.022, $p_{corr}$ = $9.855^{-05}$ and β = -0.014, $p_{corr}$ = 0.01), and higher FA in six of 15 individual WM tracts (effect sizes ranging from β = -0.013 to β = -0.021) (Fig 2 and S3.1 Results in S1 File). No MD measures were associated with seasonality.

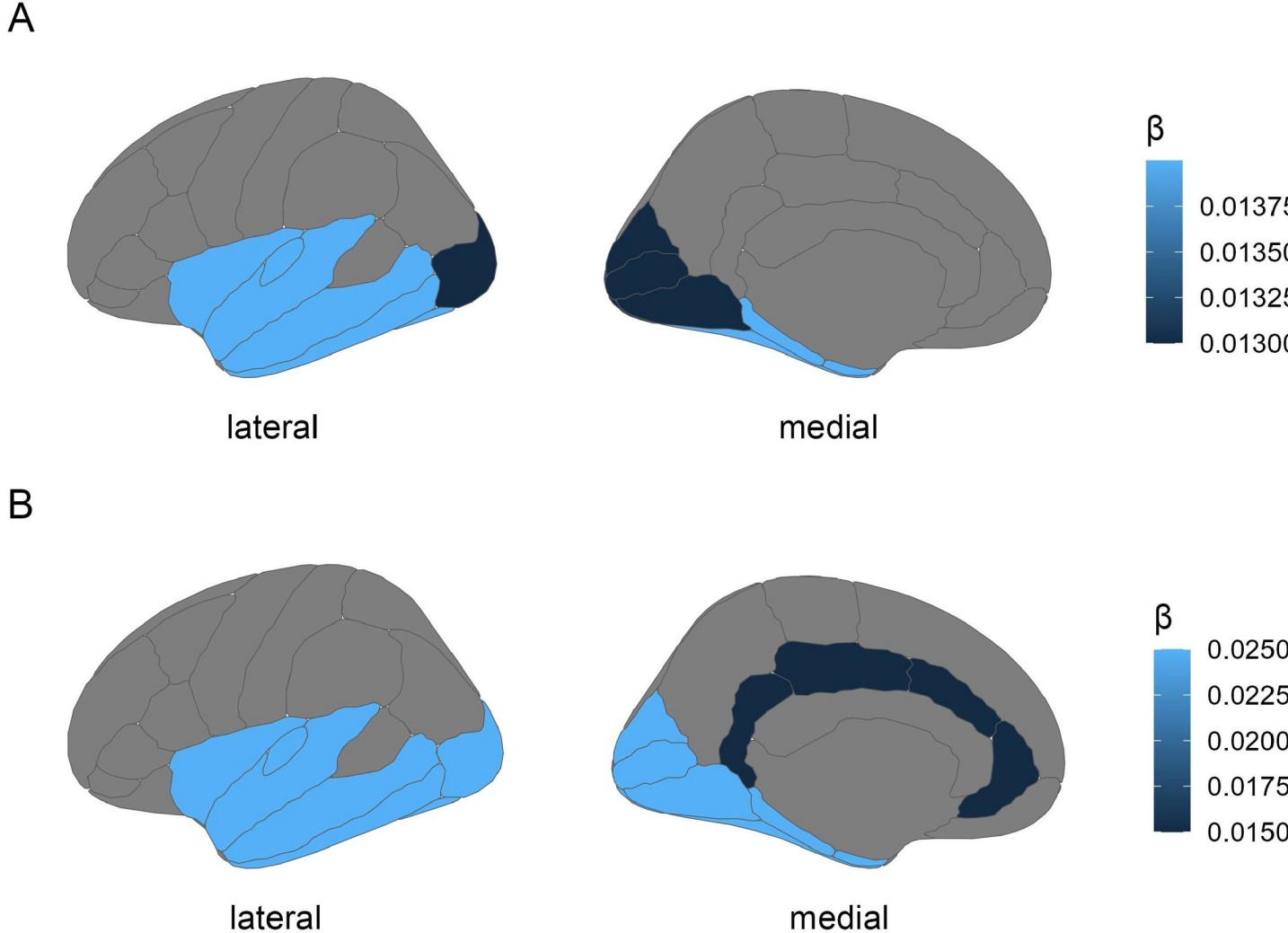

**Fig 1.** Standardised effect sizes of brain morphology measures associated with seasonality mapped onto the Desikan-Killiany-Tourville atlas for (A) mean cortical thickness of the temporal and occipital lobes in the main analyses and (B) mean cortical thickness of the temporal, occipital and cingulate lobes in the sensitivity analyses. Darker colour designates greater effect size for associations with summer births.

When birth weight was additionally covaried for in the sensitivity analyses, one additional cortical morphometric measure was identified as associated with seasonality: cingulate CT ($\beta$ = 0.015, $p_{corr}$ = 0.037) (Fig 1). With regard to WM measures, an additional association of FA in the forceps minor with winter births was identified when correcting for birth weight ($\beta$ = -0.015, $p_{corr}$ = 0.044), while all previous associations remained significant with larger effect sizes (see Fig 2, S3.2.5-S3.2.7 Results in S1 File for further details). Additionally, volume of the amygdala was associated with summer births ($\beta$ = 0.021, $p_{corr}$ = 0.042) (See S3.2.4 Results in S1 File).

## Mental health trait associations with brain imaging measures

There were no observed differences between summer P-MDD and winter P-MDD births across all brain imaging measures (See S3.5 Results in S1 File). Furthermore, there were no significant interactions between any of the mental health traits and seasonality (See S3.6.5 Results

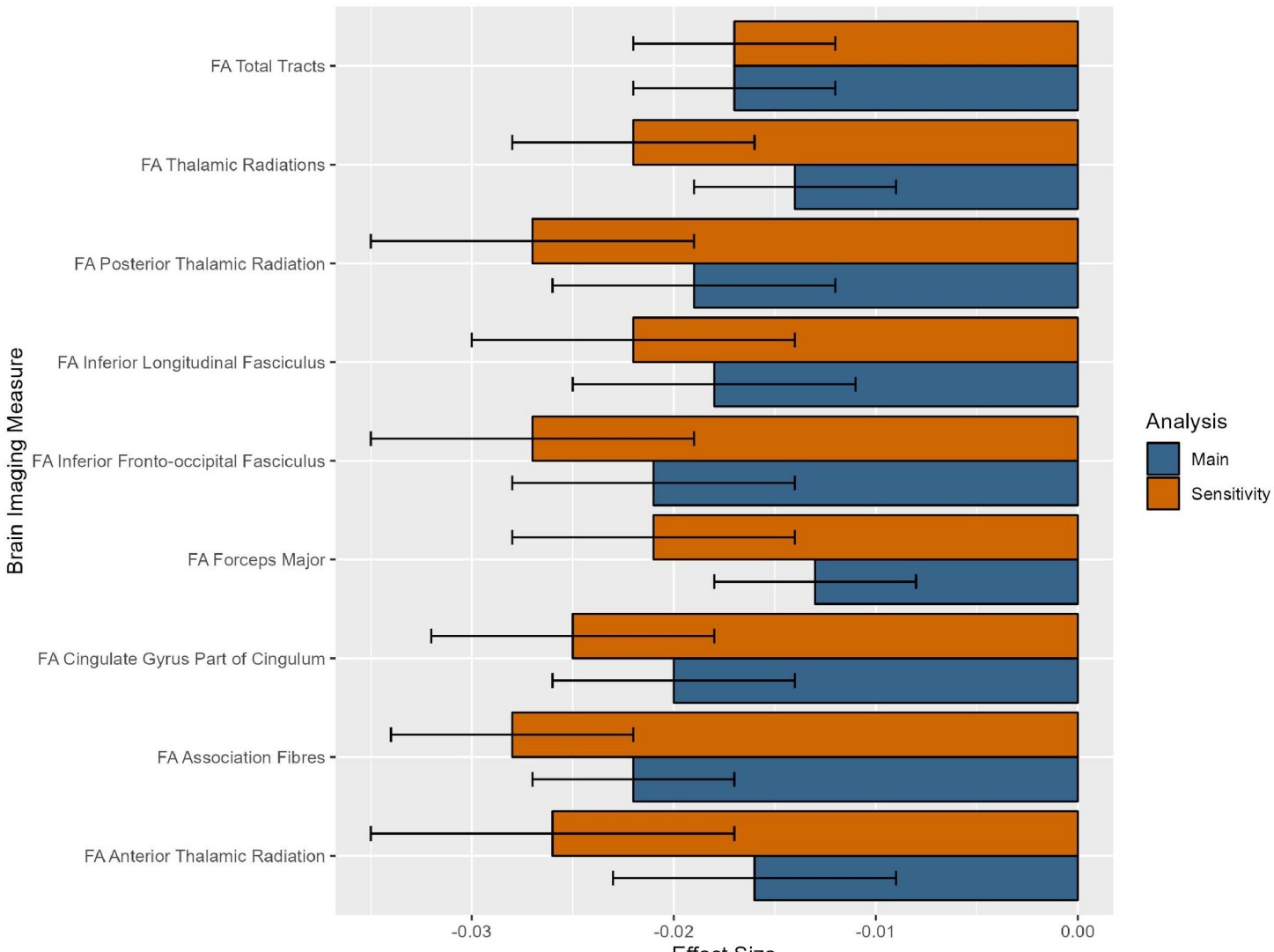

**Fig 2. Standardised effect sizes for white matter microstructure neuroimaging measures significantly associated with seasonality (p_corr<0.05 for regional/individual measures, p<0.05 for global measures) in the main analysis and sensitivity analysis.** A negative effect size for seasonality in the main and sensitivity analyses represents an increase in FA for winter births compared to summer births.

in S1 File). Individually, both P-RMDD and P-SEMDD were associated with increases in gMD (β = 0.059 and 0.045, $p = <0.05$) and decreases in gFA (β = -0.058 and -0.049, $p = <0.001$) with the effect being more marked for P-RMDD (Fig 3). (See S3.4.5 Results in S1 File).

Regional WM microstructure associations with P-RMDD were observed for lower FA in gTR (β = -0.090, $p_{corr} = <0.0001$) and gAF (β = -0.058, $p_{corr} = <0.05$) and higher MD in gTR (β = 0.072, $p_{corr} = <0.0001$), gAF (β = 0.039, $p_{corr} = <0.05$) and gPF (β = 0.068, $p_{corr} = <0.001$) (Fig 4 and S3.4.6 Results in S1 File). P-BD was also associated with lower FA in gTR (β =

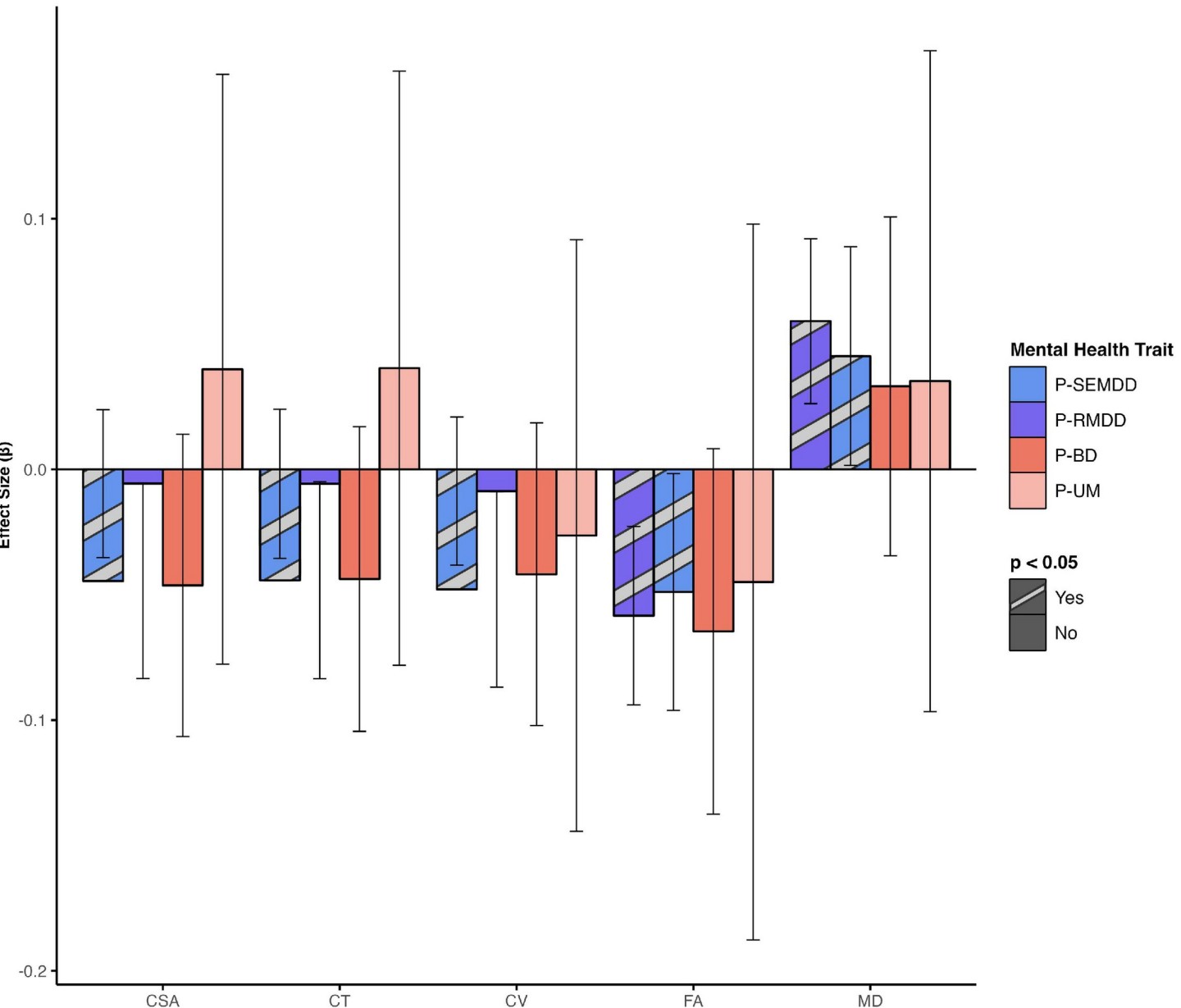

**Fig 3. Associations between mental health traits and global white matter microstructure and brain morphology measures.** P-RMDD = Probable Recurrent Major Depressive Disorder; P-SEMDD = Probable Single episode Major Depressive Disorder; P-BD = Probable Bipolar Disorder; P-UM = Probable Unipolar Mania; CSA = cortical surface area; CV = cortical volume; FA = fractional anisotropy; MD = mean diffusivity. Each bar represents β coefficients for associations with global measures. Striped bars signify significant associations (p<0.05). Error bars represent the 95% confidence interval of the estimated coefficient.

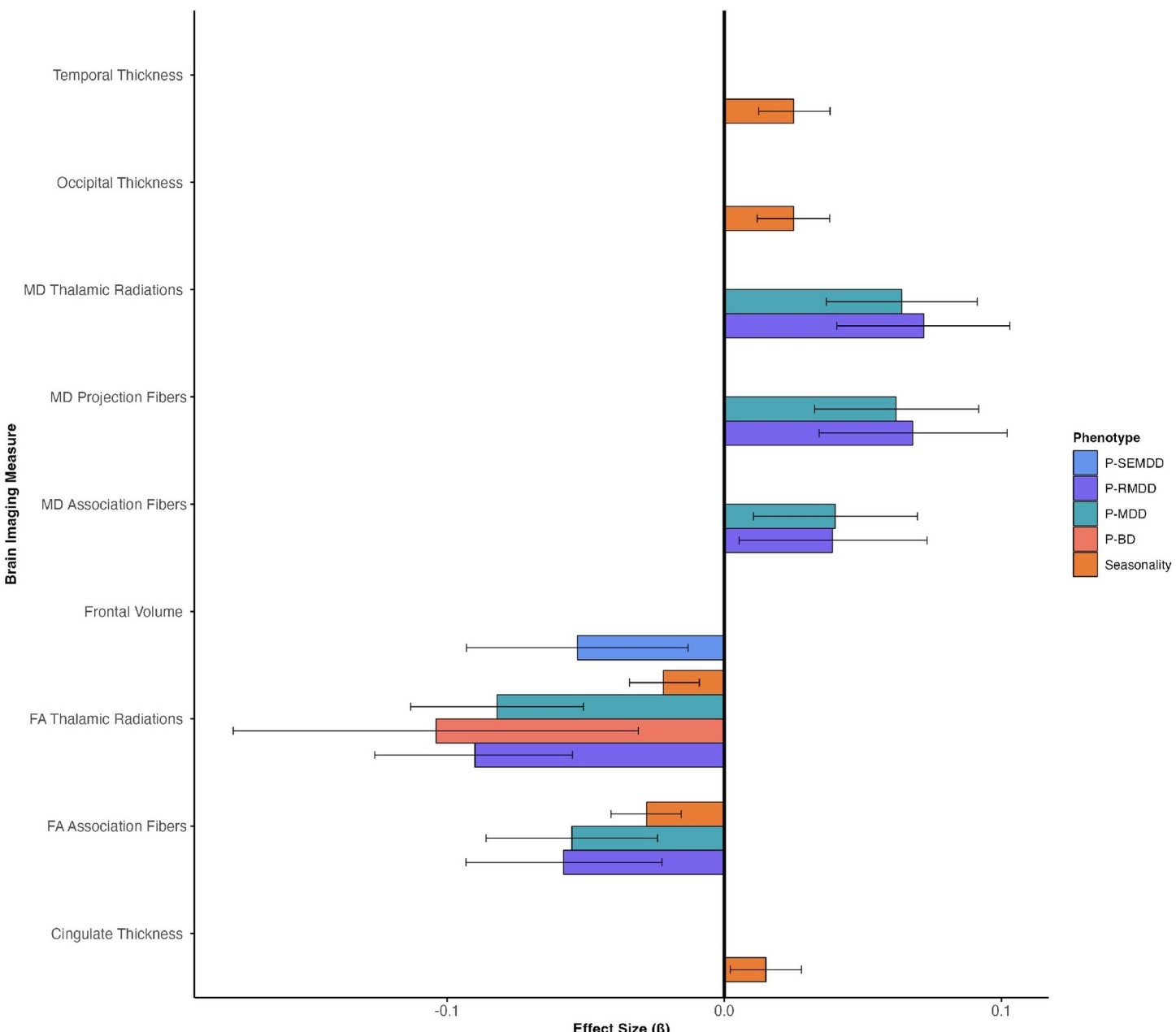

**Fig 4. Significant associations ($p_{corr}$<0.05) between mental health traits and seasonality for regional white matter microstructure and brain morphology measures.** Effect sizes for seasonality associations are from the sensitivity analysis. P-RMDD = Probable Recurrent Major Depressive Disorder; P-SEMDD = Probable Single episode Major Depressive Disorder; P-BD = Probable Bipolar Disorder. Each bar represents β coefficients for associations with regional measures. Error bars represent the 95% confidence interval of the estimated coefficient.

-0.053, $p_{corr}$ = <0.047) (Fig 4). Decreased frontal lobe CV was associated with P-SEMDD (β = -0.104, $p_{corr}$ = <0.016) (Fig 4 and S3.4.2 Results in S1 File).

There were no associations between seasonality and brain imaging measures when adjusted for mental health traits (See S3.7 Results in S1 File). The direction of effect for associations across all analyses is available in Supplementary Information (S4 Results in S1 File). Extended results are available in S5 Results, S7 and S8 Figs in S1 File.

## Discussion

This study investigated associations between seasonality, mental health traits and neuroimaging measures in a large-scale cross-sectional dataset. Seasonality was associated with P-RMDD, as well as with a range of brain morphology and white matter microstructure brain imaging measures. Summer births were associated with a higher prevalence of P-RMDD, as well as greater CT in the temporal, occipital and cingulate lobes and greater SV of the amygdala. Winter births were associated with more constrained water molecule diffusion and thus higher white matter integrity globally, in two white matter fiber tract bundles and in seven individual white matter tracts. P-RMDD was associated with decreased global, regional and individual WM integrity and decreased thalamic SV. P-SEMDD was associated with decreased global and regional WM integrity to a lesser degree as well as decreased brain morphology across all global measures and decreased CSA in the frontal lobe. P-BD was associated with reduced FA in the gTR and posterior thalamic radiations. A single significant seasonality interaction was observed with P-BD for reduced amygdala volume.

These results cautiously lend additional support to existing evidence [6,52,53] of season of birth effects in adult health in the context of a modest sized cross-sectional sample of a healthy UK-based population. They also support previous evidence of globally [39,54]and regionally [29] reduced WM integrity across a range of depression phenotypes, tract-specific reductions in recurrent MDD [55] and BD [56,57], as well as brain morphology changes in MDD [58–60]. They do not, however, fully characterise the relationship between lifetime mental health traits, brain imaging measures and seasonality of birth.

In this study P-RMDD was associated with summer births, aligning with northern hemisphere spring associations with MDD risk in English outpatients (*n* = 16,726) [6], severity in a small MDD case-control study (45 cases and 90 controls) [52] and earlier disease onset in 855 Korean MDD patients [53]. However, there exist idiosyncrasies in the seasonality phenotypes derived in these studies, which range from monthly, to two- and four-season seasonality categories per participant, making for an inexact comparison. A hemispheric six-month shift has also been observed between seasonality and depression symptoms, with higher scores associating with spring births (March-May) in the Northern hemisphere and autumn births (September-November) in the Southern hemisphere, in a small young adult and adolescent cohort [61]. Although an excess of August births in MDD patients has also been observed in the Southern hemisphere within a Brazilian retrospective study of MDD cases and controls (*n* = 98,457) [62]. However, no associations between month of birth and depression symptoms were found in a similarly aged cohort in a recent Europe wide study (*n* = 72,370) [63]. For P-RMDD, therefore, our findings provide a seasonality association within a UK population sample adding another datapoint to the ambiguous literature.

Unlike P-RMDD, single episode MDD was not associated with seasonality in this study. The more than two-fold reduction in sample size for P-SEMDD (*n* = 13,721) may have limited statistical power to detect associations. However, an unshared aetiology may also be a factor, with more severe neurophysiological observations made in clinically comparable recurrent MDD cases versus single episode MDD cases [64], and an earlier age of onset and familial risk for recurrent MDD [65,66]. Gene  environment effects [67] and genetic variants [68,69] associated with recurrent MDD support this distinction. Therefore, seasonality effects may vary within depressive subtypes, with a more marked effect on persistent cases in response to seasonal perinatal and natal environments. Ultimately, the discrepancy in findings for seasonality associations with recurrent and single episode MDD could be better addressed within a clinical sample, given factors such as the recall bias involved in enumerating lifetime depressive

episode measures within UK Biobank [34], and the overall reliance on self-reported measures to distinguish these diagnoses.

Bipolar affective disorder has been associated with January births [6] (OR = 1.09, 95% CI = 1.03–1.15, $p$ = 0.002), and for DSM-III bipolar disorder cases a seasonal pattern has been observed with significant excess births in December and a total of 5.8% seasonal excess births to expected births [70]. However, the current study did not find a seasonal association, possibly due to the inclusion of both P-RMDD and P-SEMDD within P-BD since only the former associated with seasonality. It is also possible that the self-reported phenotype used may not fully capture cases with sufficient fidelity to yield associations. UM, a relatively unexplored BD-subtype [71], had the largest effect size of the mental health traits but was not associated with seasonality potentially due to the small number of cases available. Re-examination of this phenotype in a larger UM sample is therefore warranted.

In line with previous studies, we find seasonality to be associated with a cluster of white matter microstructure measures, with novel findings for brain morphology regional measures. Overall, three regional measures were associated with seasonality, all of which were in the mean thickness category, despite the lack of associations for volumetric measures and seasonality expressed as daylength in similarly powered UKB studies [16]. Thickness of the temporal lobe has been shown to be significantly reduced in schizophrenic patients [72], and the disorder's own association with winter births [73], may offer an opportunity for further study.

The observation of multiple negative associations for FA measures in association with summer births, corroborates previous findings in which sections of the corpus callosum, the internal capsule, the corona radiata, the posterior thalamic radiation and the sagittal striatum were found to have decreased FA values in summer compared to winter births [74]. For summer births, lower FA measures, or less restricted and more isotropic diffusivity, may point to greater tissue disorganisation, itself generally accompanied by reduced axonal myelination, axonal loss, or a higher proportion of crossing fibers. Since global and regional measures yielded the strongest associations, alongside seven individual measures, seasonality may exert a non-localised effect on white matter integrity and thus warrant further analysis to specify possible mechanisms. The overall association of the thalamic radiations bundle and the association fibers bundle, composed of two and three individual tracts with individual associations respectively, provides novel evidence for lower values in summer births.

Lastly, although amygdala volume has not previously been associated with summer births as it was in his study, seasonal fluctuations in amygdala volume measures, with increases in the summertime and during longer photoperiods, have been found in UKB [75]. The biological underpinnings for these shifts are unknown but hormonal mediation, specifically via melatonin, is one proposed mechanism due to its role in photoperiod-induced adaptations, alongside the presence of melatonin receptors in the amygdala. It is, however, unclear at this stage whether perinatal photoperiodicity differentially affects grey matter development and if these effects are stable through development and into adulthood.

Birth seasonality effects, therefore, may independently induce penetrable changes in white matter integrity in a subset of tract bundles, an effect discernible through DTI-ascertained measures. Brain structure and connectivity measures are a promising endophenotype for mental, neurological, and physiological illness. The range of neuroimaging associations with seasonality found here offers a starting point for further probing into the mechanistic relationship between them. These findings, however, do not hold when adjusted for mental health traits. However, a notably reduced sample size (~71% less participants) for models examining both mental health traits and brain imaging measures could have incurred a loss of power to detect associations. Given the adverse effects of mental health disorders on the brain, the presence of

a probable mental health trait is also likely to be a stronger predictor of brain imaging measures than seasonality.

Global reductions in FA alongside increases in MD have been associated with self-reported, probable, and recurrent depression in UKB [39] with reductions in global FA also observed within a small ENIGMA sample of MDD patients ($n^{cases}$ = 921, $n^{controls}$ = 1265) [54]. There is also compelling evidence that regional FA and MD measures associate with overall depression, with previous studies reporting lower FA in the association fibers and thalamic radiations for self-reported, probable, recurrent, and clinical depression [39], and for both principal MDD and recurrent MDD [29]. Likewise, increased MD in the thalamic radiations and projection fibers have been associated across the same range of depression phenotypes [39], with higher MD in the association fibers also associated with self-reported depression. MDD associations with reduced FA in a number of individual tracts have also been reported in UKB, notably in the left superior longitudinal fasciculus, and bi-hemispheric superior thalamic radiation [29], posterior thalamic radiations and forceps minor [39], akin to the findings in this study, with 16 out of 25 individual WM tracts examined also found to associate with MDD patients in a large ENIGMA sample [55]. The lack of observed differences in WM microstructural measures between P-RMDD and P-SEMDD cases, was also reported in an ENIGMA study, as well as similar lack of findings for associations between MDD and brain morphometry measures [54].

Regardless of seasonal MDD trends, this study supports previous evidence of a general association between depression and reduced WM integrity. For BD, although both increased and decreased FA measures have been reported, a review of WM microstructural changes in the disorder also point to mostly FA decreases in the thalamic radiations [56], with a region of interest study reporting reduced FA in the posterior thalamic radiation for both BD patients and unaffected siblings [57]. Morphologically, we find a reduction in thalamic volume for P-RMDD cases, a finding previously reported in the largest study to date [58] examining sub-cortical volumes ($n^{cases}$ = 142) for current MDD patients compared to healthy controls, in a meta-analysis of 143 studies [59] and in a study of ~600 community-dwelling lifetime MDD participants [76], whilst left hemispheric reductions in thalamic volume was also associated with MDD in a smaller study ($n^{cases}$ = 30) [60]. Given the role of the thalamus in sensory information relay as well as various modes of higher order executive and cognition functioning [77], volumetric reductions here could be relevant to the pathology of MDD, with functional imaging abnormalities for MDD patients in this structure also having been reported [78].

## Limitations

This study was limited in geographical scope with the aim of keeping latitudinal and longitudinal variation minimal between subjects. Since season of birth effects have been shown to be greater at higher latitudes [73], possibly mediated by larger annual photoperiod shifts, studies in these regions, or meta-studies encompassing them might provide further insight.

Although our mental health trait phenotypes align well with DSM-5 diagnostic criteria [34], they do not reflect formal diagnoses and therefore may under- or over-extend seasonality associations present under stricter definitions. A symptom-by-symptom study could also elucidate individual patterns in associations, such as MDD sleep aberrances and evidenced seasonality mediated sleep differences [16]. The natural patterns of distribution in birth seasonality [79] with April and May annual birth-rate peaks in UKB [16] should also be accounted for, as should the seasonal pattern in procreation habits observed in psychiatric disorders such as schizophrenia, which may be tied to heritable components [80] of mental health disorders independently. Here, birth seasonality variation was assessed comparing winter births with summer births. However, this provides those born in spring and autumn with similar

phenotypic scores and therefore will not model differences between those born in those periods. Future studies could co-model a range of seasonality phenotypes to contextualise the extent of these potential differences.

Since historically, seasonality effects on psychiatric conditions have been more pronounced in public versus private hospitals [81] and affected by urbanicity and education level, studies that account for these variables could provide a better picture of seasonality associations, although TDI may be a sufficient proxy for these measures. Generally, the lag between the time of birth and seasonality-related outcomes in later life makes their association prone to confounders which must be further considered. Overall, the small effect sizes found in this study suggest a limited role for seasonality in adult health, although usage of larger sample sizes and inclusion of more covariates could modify this.

Lastly, although our study supports an association between seasonality and adult health, identification of the mechanisms by which these effects are actualised is beyond the scope of the current study. Longitudinal studies tracking changes in mental health and neuroimaging measures would more precisely quantify within-individual shifts over the lifetime and ease the identification of key developmental periods in which these take place. Since circadian patterns of gene expression have been demonstrated to be weaker in post-mortem human subjects with MDD [82], further studies examining deficiencies in circadian rhythmicity at the transcriptomic level and their associations with mental health traits in the context of month of birth could also be beneficial. A combinatorial approach including genetic and gene expression data would give insight into differential seasonality programming and begin to specify possible biological pathways.

In summary, we demonstrated both seasonality of birth associations with adult health as measured by mental health traits and neuroimaging measures, as well as associations between mental health traits and neuroimaging measures. Although the reported association between seasonality and P-RMDD could be specific to the parametrizations of this study and those of UK Biobank cohort, we are hopeful that they may spur further research. A continuation of the examination of seasonality associations with a wide range of mental health traits is encouraged within higher powered studies or those utilising clinical cases, especially given the limited number of interactions between mental health traits and seasonality found here. Overall, the small effect sizes of all our associations with global, regional, and individual brain imaging measures, as well as probable recurrent Major Depressive Disorder warrant replication in larger and more diverse datasets as well as those offering wider latitudinal ranges.

## Supporting information

**S1 File.**
(DOCX)

## Acknowledgments

This research was conducted using the UK Biobank resource, application number 4844. The UK Biobank study was conducted under generic approval from the NHS National Research Ethics Service (approval letter dated June 17, 2011, Ref 11/NW/0382). All participants gave full informed written consent. We would also like to thank the UK Biobank team and all the participants for their collaboration and C.A Wyse for her valuable input.

## Author Contributions

**Conceptualization:** Maria Viejo-Romero, David M. Howard.

**Data curation:** Xueyi Shen, Aleks Stolicyn.

**Formal analysis:** Maria Viejo-Romero.

**Funding acquisition:** David M. Howard.

**Investigation:** Maria Viejo-Romero, Heather C. Whalley.

**Methodology:** Maria Viejo-Romero, Heather C. Whalley.

**Software:** Xueyi Shen, Aleks Stolicyn.

**Supervision:** Heather C. Whalley, Daniel J. Smith, David M. Howard.

**Visualization:** Maria Viejo-Romero, David M. Howard.

**Writing – original draft:** Maria Viejo-Romero, David M. Howard.

**Writing – review & editing:** Maria Viejo-Romero, Heather C. Whalley, Xueyi Shen, Aleks Stolicyn, Daniel J. Smith, David M. Howard.

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
