## [Decision Letter · Decision Letter 0]

11 Oct 2023

PONE-D-23-25155An epidemiological study of season of birth, mental health, and neuroimaging in the UK BiobankPLOS ONE

Dear Dr. Howard,

Thank you for submitting your manuscript to PLOS ONE. After careful consideration, we feel that it has merit but does not fully meet PLOS ONE’s publication criteria as it currently stands. Therefore, we invite you to submit a revised version of the manuscript that addresses the points raised during the review process.

We look forward to receiving your revised manuscript.

Kind regards,

Kenji Tanigaki, Ph.D., M.D.

Academic Editor

PLOS ONE

Reviewers' comments:

Reviewer's Responses to Questions

**Comments to the Author**

1. Is the manuscript technically sound, and do the data support the conclusions?

Reviewer #1: Partly

Reviewer #2: Partly

2. Has the statistical analysis been performed appropriately and rigorously? 

Reviewer #1: I Don't Know

Reviewer #2: No

3. Have the authors made all data underlying the findings in their manuscript fully available?

Reviewer #1: Yes

Reviewer #2: Yes

4. Is the manuscript presented in an intelligible fashion and written in standard English?

Reviewer #1: Yes

Reviewer #2: Yes

5. Review Comments to the Author

Reviewer #1: I have read with interest the manuscript by Viejo-Romero et al. on the effect of season of birth on mental health in adults. It is especially relevant the fact that they use a large and high-quality database as the UK Biobank to study this effect, which has received moderate attention in recent years. Authors find small effects relating probable recurrent Major Depressive Disorder with the summer. However they do not find a similar effect with probable hypomania, mania or, importantly, single episode Major Depressive Disorder. Additionally, they find a high number of neuroimaging correlations.

Overall, reporting is of high quality and follows current standards. However, I have some issues with the article as it stands that question the robustness of its results and if solved would probably improve the paper and its addition to the field.

I have to say that I am slightly biased against the hypothesis that authors propose, as I do not think that there is a strong biological basis for any long-last consequences of date of birth. The article does not help against my skepticism of the field. All in all, the authors are not able to convincingly proof whether there is something to be learnt from the current study or it is simply an statistical fluke.

Regarding the neuroimaging data, the relevance of its results is hard to assess. As it is frequent in the neuroimaging literature, some correlations are found without any possibility to judge whether they are relevant or not… I cannot judge this section as one is easily lost in the jungle of measures and RoIs… I would however recommend the authors to include a paragraph on how the FDR correction holds when it is applied to many different comparisons independently.

The models directly linking date of birth and risk of mental health issues are slightly more interpretable. However, they are far from being straightforward. Indeed the phenotypic score for season (“The birth month of each participant was transformed via a cos function, with the lowest phenotypic score (-1) corresponding to those born in December and the highest phenotypic score corresponding to those born in June (+1)”) is only one of multiple parametrizations, which is selected without justification. Even more importantly, associations between the four mental health phenotypes and seasonality were tested using a complex logistic binomial regression analysis in which sex, age, age2, Townsend Deprivation Index, assessment centre attended and place of birth location were included as covariates. There is no description on how authors arrived at the selection of these factors. My suspicion is that the result found may not be robust to changes in the proposed model and may be the result of the “garden of forking paths” and the researcher’s degrees of freedom in their design. The following actions could provide additional confidence in the results:

1-Proof indicating that these choices were carried out a-priori (eg: a public protocol or previous references by the authors or other researchers using the exact same parametrizations and models). If not possible, at least some information on how and why they were selected.

2-A declaration by the authors indicating that this was the only model tested.

3-Some sort of multiverse testing or robustness checks showing how different changes in the analytical decisions affect the results. An easy one would be to show the results of the logistic model with different combinations of the covariates.

In addition to the above, the article as written gives too little weight to the uncertainty present in the results. Some things that could be improved:

1-include in abstract the negative results with the other disorders. If authors need to cut some parts, the imaging info in the abstract could be mostly dropped.

2-Uncertainty of the results should be included, for example in P13-L246, but also other parts of the discussion.

3-Reduce Post-hoc reasoning on why negative results were found: reasons provided for example in p14 l 262 are probably less likely than the elephant in the room (that is, that the positive results found was simply an statistical fluke).

4-Include a section on limitations of the current study, which should include the fact that there is not a good explanation on why results where only found for recurrent MDD.

Reviewer #2: UK Biobank data are used to test the hypothesis that seasonality influences the incidence of affective disorders and brain structural measures derived from MRI. An excess number of diagnoses of recurrent depression is found during the summer months and seasonality is associated with some neuroimaging markers.

Whilst the motivation for the conception and design is broadly sound, the exclusion of non-white participants without any scientific justification, or indeed any justification at all, is unacceptable. Whilst I am in no doubt that the authors hold no ill intentions, funding agencies – including the Wellcome Trust – are absolutely clear that scientific studies should be for the benefit of everyone. Moreover, we are ethically compelled to undertake research for the whole community unless there are specific reasons not to do so. This is particularly relevant here given that in the UK people with Black ethnicity have almost twice the prevalence of depression compared to European (white) ethnicity [doi: 10.1017/S0033291714002967].

The article requires careful reconsideration. The exclusion of any group based on their ethnicity needs strong and credible justification.

6. PLOS authors have the option to publish the peer review history of their article (what does this mean?). If published, this will include your full peer review and any attached files.

Reviewer #1: No

Reviewer #2: No

---

## [Author Response · Author response to Decision Letter 0]

23 Nov 2023

We have uploaded a response to reviewers document providing a point-by-point response to all comments raised.

---

## [Decision Letter · Decision Letter 1]

15 Dec 2023

PONE-D-23-25155R1An epidemiological study of season of birth, mental health, and neuroimaging in the UK BiobankPLOS ONE

Dear Dr. Howard,

Thank you for submitting your manuscript to PLOS ONE. After careful consideration, we feel that it has merit but does not fully meet PLOS ONE’s publication criteria as it currently stands. Therefore, we invite you to submit a revised version of the manuscript that addresses the points raised during the review process.

We look forward to receiving your revised manuscript.

Kind regards,

Kenji Tanigaki, Ph.D., M.D.

Academic Editor

PLOS ONE

Reviewers' comments:

Reviewer's Responses to Questions

**Comments to the Author**

1. If the authors have adequately addressed your comments raised in a previous round of review and you feel that this manuscript is now acceptable for publication, you may indicate that here to bypass the “Comments to the Author” section, enter your conflict of interest statement in the “Confidential to Editor” section, and submit your "Accept" recommendation.

Reviewer #1: All comments have been addressed

Reviewer #2: (No Response)

2. Is the manuscript technically sound, and do the data support the conclusions?

Reviewer #1: Yes

Reviewer #2: No

3. Has the statistical analysis been performed appropriately and rigorously? 

Reviewer #1: Yes

Reviewer #2: Yes

4. Have the authors made all data underlying the findings in their manuscript fully available?

Reviewer #1: Yes

Reviewer #2: Yes

5. Is the manuscript presented in an intelligible fashion and written in standard English?

Reviewer #1: Yes

Reviewer #2: Yes

6. Review Comments to the Author

Reviewer #1: While I am still biased against the study results, authors have shown an extreme responsiveness to my comments and have provided as much evidence of the reliability of their results as it was feasible to ask. Hence, I believe that the article in its current form has greatly improved and is suitable for publication. I want to congratulate authors for their effort in their updated version.

Reviewer #2: UKBiobank data were used to investigate seasonality of birth with mental health traits and brain morphology and microstructure measured from MRI. Depressive traits were associated with summer births as were significant increases in cortical thickness. Changes to diffusion measures were associated with winter births.

The scientific reasoning behind this study described in the Introduction is that the season of birth affects brain structure and changes to brain structure are observed in mental health disorders.

The observed effects of seasonality of birth are weak and do not help clarify a confused literature and seasonality of birth effects include some areas that overlap with a recent meta-analysis of cortical thickness in cross-sectional primary literature of (diagnosed) MDD, although the direction is reversed, although the entire cohort is used in the linear model rather than focusing on any particular trait.

A more direct comparison of the underlying hypothesis would be that once seasonality of birth and prevalence is established for, as it would appear, MDD traits, then first undertake a case-control comparison of all participants with MDD traits, and then a MDD traits summer compared to winter births. The expectation would be a pattern of differences similar to that seen in the literature with effect sizes increased in summer compared to winter. Alternatively, apply the linear model to predict brain imaging measures with a main effect of mental health trait and a trait x month interaction, expecting effects in brain regions that match to some extent with the literature.

In short, a credible hypothesis is posited that should be directly addressed by the analysis strategy.

7. PLOS authors have the option to publish the peer review history of their article (what does this mean?). If published, this will include your full peer review and any attached files.

Reviewer #1: No

Reviewer #2: No

---

## [Author Response · Author response to Decision Letter 1]

12 Feb 2024

Dear Editor,

We thank you once again for the opportunity provided to re-submit our manuscript entitled “An epidemiological study of season of birth, mental health, and neuroimaging in the UK Biobank” (PONE-D-23-25155) to PLOS ONE. We provide further answers to Reviewers 2’s comments below.

Reviewer #1: While I am still biased against the study results, authors have shown an extreme responsiveness to my comments and have provided as much evidence of the reliability of their results as it was feasible to ask. Hence, I believe that the article in its current form has greatly improved and is suitable for publication. I want to congratulate authors for their effort in their updated version.

Reviewer #2: UKBiobank data were used to investigate seasonality of birth with mental health traits and brain morphology and microstructure measured from MRI. Depressive traits were associated with summer births as were significant increases in cortical thickness. Changes to diffusion measures were associated with winter births. The scientific reasoning behind this study described in the Introduction is that the season of birth affects brain structure and changes to brain structure are observed in mental health disorders.

The observed effects of seasonality of birth are weak and do not help clarify a confused literature and seasonality of birth effects include some areas that overlap with a recent meta-analysis of cortical thickness in cross-sectional primary literature of (diagnosed) MDD, although the direction is reversed, although the entire cohort is used in the linear model rather than focusing on any particular trait.

1. A more direct comparison of the underlying hypothesis would be that once seasonality of birth and prevalence is established for, as it would appear, MDD traits, then first undertake a case-control comparison of all participants with MDD traits and then a MDD traits summer compared to winter births. The expectation would be a pattern of differences similar to that seen in the literature with effect sizes increased in summer compared to winter.

We thank the reviewer for this comment. 

(a) A case-control comparison of all participants with MDD traits was conducted for all brain imaging measures and the methods for this have now been added to the manuscript (p12, lines 209-213) and below:

“First, models to examine associations between brain imaging measures and probable Major Depressive Disorder cases (P-MDD) were constructed. P-MDD included all participants who had been previously classified as P-RMDD or P-SEMDD cases, with P-MDD controls defined as those participants who were controls for both P-RMDD or P-SEMDD (See S2.5.1 for demographic table).”.

(b) This was supplemented by a case-control comparison for RMDD cases vs SEMDD cases for all brain imaging measures based on the differences we observed when they were examined separately. The methods for this analysis can also now be found in the manuscript (p12, lines 216-219) and below: 

“Thirdly, additional models were constructed to investigate associations between brain imaging measures and P-RMDD cases compared to P-SEMDD cases (See S2.5.3 for demographic table). All covariates were kept consistent with the main analysis.”.

(c) Finally, the case/control comparison of summer born MDD cases compared to winter born MDD cases for all brain imaging measures was also conducted with methods reported in (p12, lines 213-216) and below:

“Secondly, models were constructed to explore associations between brain imaging measures and winter birth P-MDD cases compared to summer birth P-MDD cases. Seasonality was split into winter births (December, January, February) and summer births (June, July, August) and coded as a categorical variable (See S2.5.2 for demographic table).”.

2. Alternatively, apply the linear model to predict brain imaging measures with a main effect of mental health trait and a trait x month interaction, expecting effects in brain regions that match to some extent with the literature. In short, a credible hypothesis is posited that should be directly addressed by the analysis strategy.

We thank the reviewer for this comment. We also took on board this suggestion and have now supplemented the methods and results for this analysis in (p12-13, lines 219-225) and below. Please note, to ensure consistency with the main analysis, the “month interaction” was replaced with a “seasonality” interaction.

“Lastly, models were constructed to investigate associations between mental health traits and neuroimaging measures in the context of seasonality. For each neuroimaging measure a linear/fixed effect model was applied, retaining the same covariates as in the main analysis plus one mental health trait (n=4: P-RMDD, P-SEMDD, P-BD and P-UM) and an interaction term between the mental health trait and seasonality. As in the main analysis, mixed-effects models were used for bi-hemispheric measures. The same multiple correction strategy was applied as in the main analysis.”.

The results for the above analyses (both comments 1 and 2) can now also be found combined into pages 14-15 (lines 263-294) with multiple figures (Figures 3-4, S7.Fig and S8.Fig) also made available. Discussion of these analyses can be found on lines 302-307, 308-314, and 381-416. The abstract and conclusion section have also been lightly modified to reflect these new results. Full results for additional analyses are also available in the Supplementary materials (S2.5, S3.4-7, S4) with extended results in S5 and supplemental figures (S7 Fig and S8 Fig).

---

## [Decision Letter · Decision Letter 2]

28 Feb 2024

An epidemiological study of season of birth, mental health, and neuroimaging in the UK Biobank

PONE-D-23-25155R2

Dear Dr. Howard,

We’re pleased to inform you that your manuscript has been judged scientifically suitable for publication and will be formally accepted for publication once it meets all outstanding technical requirements.

Kind regards,

Kenji Tanigaki, Ph.D., M.D.

Academic Editor

PLOS ONE

Additional Editor Comments (optional):

Reviewers' comments:

Reviewer's Responses to Questions

**Comments to the Author**

1. If the authors have adequately addressed your comments raised in a previous round of review and you feel that this manuscript is now acceptable for publication, you may indicate that here to bypass the “Comments to the Author” section, enter your conflict of interest statement in the “Confidential to Editor” section, and submit your "Accept" recommendation.

Reviewer #2: All comments have been addressed

2. Is the manuscript technically sound, and do the data support the conclusions?

Reviewer #2: Yes

3. Has the statistical analysis been performed appropriately and rigorously? 

Reviewer #2: Yes

4. Have the authors made all data underlying the findings in their manuscript fully available?

Reviewer #2: Yes

5. Is the manuscript presented in an intelligible fashion and written in standard English?

Reviewer #2: Yes

6. Review Comments to the Author

Reviewer #2: (No Response)

7. PLOS authors have the option to publish the peer review history of their article (what does this mean?). If published, this will include your full peer review and any attached files.

Reviewer #2: **Yes: **John Suckling

---

## [Editor Report · Acceptance letter]

29 Apr 2024

PONE-D-23-25155R2 

PLOS ONE

Dear Dr. Howard, 

I'm pleased to inform you that your manuscript has been deemed suitable for publication in PLOS ONE. Congratulations! Your manuscript is now being handed over to our production team.

Kind regards, 

on behalf of

Dr. Kenji Tanigaki 

Academic Editor

PLOS ONE